Static versus dynamic muscle modelling in extinct species: a biomechanical case study of the Australopithecus afarensis pelvis and lower extremity

Wiseman Ashleigh L.A. alw96@cam.ac.uk 1
Charles James P. 2
Hutchinson John R. 3
1 McDonald Institute for Archaeological Research, University of Cambridge , Cambridge , United Kingdom
2 Evolutionary Morphology and Biomechanics Lab, Institute of Life Course and Medical Sciences, University of Liverpool , Liverpool , United Kingdom
3 Structure and Motion Laboratory, Department of Comparative Biomedical Sciences, Royal Veterinary College , Hatfield , United Kingdom
Young Jesse
Electronic publication date: 2024 Jan 31
Publication date: 2024
Volume: 12
Electronic Location ID: e16821
Received 2023 May 5; Accepted 2024 Jan 2
Copyright: ©2024 Wiseman et al.
Copyright year: 2024
Copyright holder: Wiseman et al.
License: This is an open access article distributed under the terms of the Creative Commons Attribution License, which permits unrestricted use, distribution, reproduction and adaptation in any medium and for any purpose provided that it is properly attributed. For attribution, the original author(s), title, publication source (PeerJ) and either DOI or URL of the article must be cited.
License URL: https://creativecommons.org/licenses/by/4.0/

Keywords: Hominin, Biomechanics, Hill-type muscle model, Musculoskeletal, Simulation

Funding: Leverhulme Trust Early Career Fellowship ECF-2021–054 The Isaac Newton Trust, University of Cambridge Ashleigh L.A. Wiseman was supported by a Leverhulme Trust Early Career Fellowship (grant number: ECF-2021–054) and by the Isaac Newton Trust, University of Cambridge. The funders had no role in study design, data collection and analysis, decision to publish, or preparation of the manuscript.

==============================
The force a muscle generates is dependent on muscle structure, in which fibre length, pennation angle and tendon slack length all influence force production. Muscles are not preserved in the fossil record and these parameters must be estimated when constructing a musculoskeletal model. Here, we test the capability of digitally reconstructed muscles of the Australopithecus afarensis model (specimen AL 288-1) to maintain an upright, single-support limb posture. Our aim was to ascertain the influence that different architectural estimation methods have on muscle specialisation and on the subsequent inferences that can be extrapolated about limb function. Parameters were estimated for 36 muscles in the pelvis and lower limb and seven different musculoskeletal models of AL 288-1 were produced. These parameters represented either a ‘static’ Hill-type muscle model (n = 4 variants) which only incorporated force, or instead a ‘dynamic’ Hill-type muscle model with an elastic tendon and fibres that could vary force-length-velocity properties (n = 3 variants). Each muscle’s fibre length, pennation angle, tendon slack length and maximal isometric force were calculated based upon different input variables. Static (inverse) simulations were computed in which the vertical and mediolateral ground reaction forces (GRF) were incrementally increased until limb collapse (simulation failure). All AL 288-1 variants produced somewhat similar simulated muscle activation patterns, but the maximum vertical GRF that could be exerted on a single limb was not consistent between models. Three of the four static-muscle models were unable to support >1.8 times body weight and produced models that under-performed. The dynamic-muscle models were stronger. Comparative results with a human model imply that similar muscle group activations between species are needed to sustain single-limb support at maximally applied GRFs in terms of the simplified static simulations (e.g., same walking pose) used here. This approach demonstrated the range of outputs that can be generated for a model of an extinct individual. Despite mostly comparable outputs, the models diverged mostly in terms of strength.

Introduction

Skeletal muscle (henceforth, just muscle) is a soft tissue that can be lengthened and can actively shorten, thus producing a force capable of generating movement. The maximal force of a muscle is dependent upon the cross-sectional area of its fibres and its maximal velocity is determined by the lengths of its fibres in association with the length of the tendon and the pennation angle at which the fibres insert into said tendon (e.g., Friederich & Brand, 1990; Cox et al., 2019). The architecture of a muscle (three parameters: fibre length, pennation angle, cross-sectional area) determines a muscle’s contractile force and the distance over which a muscle will contract, i.e., its ‘working range’ (Eng et al., 2008; Martin et al., 2020). These three muscle architectural parameters are commonly used in the Hill-type model of muscle contraction (Zajac, 1989). The Hill-type muscle model comprises the muscle–tendon unit in which the elastic properties of the tendon are represented by a spring in series with the contractile element (‘muscle’) and a parallel elastic element (here not discussed further; see Zajac, 1989; Millard et al., 2013).

However, muscles and other soft tissues are rarely preserved in the fossil record and instead we are left with the bare bones (Shaw, 2010). A scientist cannot simulate the locomotion of an extinct tetrapod without first reconstructing the missing soft tissues of the limbs. For example, in hominin individuals this is best achieved by first creating musculoskeletal models of extant analogous specimens whereby musculature can be directly measured and used as a comparative framework, such as models of chimpanzees (O’Neill et al., 2017; O’Neill et al., 2013; O’Neill et al., 2015; O’Neill et al., 2022; O’Neill, Nagano & Umberger, 2023) or macaques (Saito et al., 2021; Ogihara et al., 2009; Shi et al., 2012). These models can then be used to guide the creation of musculoskeletal models of hominin individuals (Wang et al., 2004; Sellers et al., 2005; Nagano et al., 2005; Wiseman, 2023; Karakostis et al., 2021; O’Neill, Nagano & Umberger, 2023).

Two musculoskeletal models of Australopithecus afarensis were published by different research teams in 2023 (Wiseman, 2023; O’Neill, Nagano & Umberger, 2023). Despite these models using distinct skeletal reconstructions (i.e., different pelvis reconstructions, one recreated by Brassey et al. (2018) modified by Wiseman (2023), versus the reconstruction by Lovejoy (1979)—differences described in detail by Wiseman (2023)) and different ways to define musculature (i.e., one approach used polygonal muscle modelling based upon MRI scans of a human to define muscle paths, whilst the other used comparative lines of action from humans and chimpanzees), the models were overall comparable in terms of moment arms (Wiseman, 2023; O’Neill, Nagano & Umberger, 2023). While further comparisons are yet to be made, the comparability is promising and suggests that different modelling assumptions have not driven divergence in modelling outputs (e.g., Demuth et al., 2023a). However, to simulate motion in these models, architectural parameters are required, which may introduce variations in outputs if the architectural parameters are estimated using different approaches (see: Charles et al., 2022; Kramer et al., 2022).

Architectural estimates in extinct taxa are typically scaled from living species, assuming that musculature is ‘phylogenetically bracketed’ (Witmer, 1995; Molnar et al., 2020; Bryant & Russell, 1992), although see Dickinson et al. (2021). Different scaling methods have been established over the last 50 years to compare how muscle limb architecture varies among extant species. For example, for mammals that are geometrically similar, homologous muscles have their masses directly proportional to body mass, their physiological cross-sectional areas proportional to body mass0.67, and muscle fibre lengths proportional to body mass0.33 (Alexander et al., 1981). In primates, studies have shown that forelimb muscle masses and fibre lengths scale with isometry (Leischner et al., 2018), and fore- and hindlimb fascicle lengths and muscle belly masses scale with positive and negative allometry respectively (Payne et al., 2006). For estimating these parameters in extinct species, some studies have scaled fibre lengths directly from extant species by body mass (Sellers et al., 2005; Nagano et al., 2005). Other studies instead assumed that all fibres were parallel and proportional to segment length, in the latter case based upon data derived from the extant phylogenetic bracket (Hutchinson, 2004a). Muscle masses have also been scaled from body mass-related proportions in samples of extant taxa (e.g., Sellers et al., 2017). Whilst scaling from one specimen to another can be straight forward, not all parameters are easily scaled, and there may also be issues with direct scaling from one specimen to another which could include intra-specific differences in body mass or proportions.

Bishop, Cuff & Hutchinson (2021a) developed a scaling method based on isometric scaling of muscle architecture which incorporates a large sample size of measured values from closely related extant taxa to estimate architectural properties in extinct taxa. A plot of normalised muscle mass and normalised fibre length was created for each homologous muscle (Bishop, Cuff & Hutchinson, 2021a). From this, two variants were produced: the arithmetic mean of the plotted points and the centroid of the plotted points. The mean of these two values was then input as the muscle mass and fibre length of the respective muscle. Whilst informative and a step forward for estimating these parameters in extinct taxa, this approach did not specifically estimate pennation angle in the extinct taxa but rather incorporated measured values into normalised muscle mass values of the extant dataset, and maximal isometric force was assumed to be proportional to body mass (i.e., uniform across all muscles).

Demuth, Wiseman & Hutchinson (2023b) expanded upon this method in their hindlimb model and simulation of locomotor biomechanics in the extinct species Euparkeria capensis. Their approach reconstructed 3D volumetric hindlimb muscles from which muscle masses were directly quantified rather than estimated using the dataset from extant taxa. Demuth, Wiseman & Hutchinson (2023b) found that the different methods of estimating muscle architecture produced mostly comparable simulated muscle activations, although when using a 3D volumetric approach, the maximum force applied to the ground far exceeded that of the other variants.

Both of these aforementioned studies (Bishop, Cuff & Hutchinson, 2021a; Demuth, Wiseman & Hutchinson, 2023b) omitted the force-length relationship of the tendon and instead assumed that the whole muscle–tendon unit behaved uniformly, but the fibres could obtain non-isometric lengths. The role of the tendon is physiologically important, but often ignored for simplicity in models of extinct species (e.g., Bishop et al., 2021b; Demuth, Wiseman & Hutchinson, 2023b). Yet tendon elasticity becomes more crucial in tendons whose slack length is longer than the optimal muscle fibre length (Yamaguchi, 2001; Zajac, 1989). The incorporation of a tendon’s slack length can help modulate the force-producing capacity of a muscle (Millard et al., 2013; Cox et al., 2019). If we want to simulate the locomotion of an extinct taxon, we first need to understand how the lack of an elastic tendon influences outputs of a simulation, which in turn can help evaluate simulation results, such as for extinct taxa that were simulated with the assumption of isometric muscle activity (Bishop et al., 2018; Bishop et al., 2021b; Demuth, Wiseman & Hutchinson, 2023b).

It is not just the biomechanics of the tendon which should be tested in simulations. Demuth, Wiseman & Hutchinson (2023b) estimated muscle parameters in three different ways and found that each method produced different fibre lengths, which was argued to be inconsequential for their study. However, differences in fibre length could influence inferences of muscle specialisations, which can be quantified using the relative ratio of fibre lengths to the muscle’s physiological cross-sectional area (PCSA) (e.g., Charles et al., 2020; Payne et al., 2006; Sharir, Milgram & Shahar, 2006). Charles et al. (2022) compared fibre length data in humans from numerous published studies and found that each study reported profoundly different fibre lengths for the same muscles. Using the ratio of fibre length to the PCSA of the muscle, Charles et al. (2022) found that these variations in length produced different muscle specialisations, for instance from velocity-driven (long fibres) to force-driven (short fibres). Whilst some previous sensitivity analyses on musculoskeletal models of dinosaurs have demonstrated that variations in fibre lengths do not influence major conclusions from musculoskeletal biomechanical analyses (e.g., Hutchinson, 2004b), other studies testing the sensitivity of muscle masses and fibre lengths in human models have instead shown that differences in architectural parameters, such as fibre lengths, can change muscle force outputs by up to 15% and variations do influence simulation outputs (Kramer et al., 2022); although the latter study did not use a Hill-type muscle model. The effects of architectural properties on modelling and simulation results have been extensively examined and reported on in the literature, especially for human models (see references cited above and those cited therein for examples).

Several different methods exist to estimate muscle parameters for extant and extinct taxa and any one of the aforementioned methods could be used to create a model of an extinct taxon. Yet, this poses the question: how replicable would a model and subsequent outputs be if different researchers created a musculoskeletal model of the same specimen (i.e., the two Australopithecus afarensis models) but estimated architectural parameters using a different approach? This is particularly pertinent for fossil specimens in which the model cannot be tested against empirical data. We need to know the variance of modelling outputs for extinct taxa prior to any complex simulations of locomotion, which can be computationally expensive and time-consuming and, under such conditions, sensitivity analyses may not be feasible. Therefore, the main aim of this study is to create several musculoskeletal models of an extinct individual each with variable architectural parameters. By systematically varying these parameters under rather simplistic simulation conditions, this study will provide insights into the range of possible outcomes.

Here, we use the hominin Australopithecus afarensis specimen AL 288-1 as an example for which seven musculoskeletal models were created. All models were identical in their creation and simulation setup, with the only variance being that each model was based upon a different estimation method for muscle parameters, following published studies’ methods (e.g., Bishop, Cuff & Hutchinson, 2021a; Demuth, Wiseman & Hutchinson, 2023b). The AL 288-1 specimen was discovered in Hadar, Ethiopia and was dated to 3.2 million years ago (Johanson et al., 1982; Kimbel, Johanson & Rak, 1994). The postcranial skeletal morphology indicates bipedality (e.g., Gruss, Gruss & Schmitt, 2017; Lovejoy, 2005; Lovejoy, 2007; Lovejoy, 1975; Ward, 2002), although there is ongoing debate about the frequency (obligate versus facultative) and effectiveness of this mode of locomotion. While a limited number of biomechanical simulations of locomotion suggest that this species was bipedal (Nagano et al., 2005; Sellers et al., 2005; Wang et al., 2004), the prevailing consensus is that AL 288-1 was an efficient upright walker. Nonetheless, certain studies propose the possibility of a crouched-like gait for this specimen (e.g., Stern & Susman, 1983). Here, we assume that this individual could use an erect-style limb posture like modern humans.

Importantly, this study does not aim to assess AL 288-1′s efficiency and capability to maintain single limb support. Instead, we use hypothetical human-like lower limb joint postures combined with each of the musculoskeletal models, in an inverse simulation with a static optimisation approach, to address our aims, as we explain below.

We used the Wiseman (2023) AL 288-1 musculoskeletal model in which 3D volumetric reconstructions of each muscle were previously reconstructed. That study made preliminary inferences of muscle functions (e.g., joint flexor/extensor) from the moment arms of each volumetric muscle and tentatively concluded that the AL 288-1 model was capable of human-like lower limb joint postures, but the model lacked muscle architectural parameters. Our approach predominantly is data exploration in which a hypothetical limb pose is modelled and divergent Au. afarensis skeletal reconstructions are ignored, such as debates surrounding pelvic reconstructions (i.e., Wiseman, 2023; Tague & Lovejoy, 1986). Our aims are to address the following questions:

1. How do the different estimation methods for muscle parameters influence maximal performance (sustainable ground reaction force), in our simplified simulation scenario?

2. Do these different methods produce muscles with different inferred architectural specialisations (force versus velocity)?

3. Do any of the methods produce a muscle which could generate relatively greater forces or operate on a different part of its force-length curve?

4. How do these methods affect simulated muscle activation patterns (i.e., different groups activated, or the same muscle to different activation levels)?

5. What are the tentative differences in muscle activations between species? We do not aim to provide a comprehensive analysis of the differences in locomotion between species. Instead, our data explorations via simulation form the preliminary groundwork for subsequent studies.

The aim of this study is not to assess outputs that can be tested against empirical data (i.e., a human model), but rather to understand the range of outputs which can be generated for a fossil specimen (i.e., AL 288-1) and to assess if such outputs are replicated when underlying model conditions are changed, thus testing the consistency of simulation results.

Materials & Methods

Musculoskeletal models

A modern human (henceforth, just ‘human’) musculoskeletal model was used to evaluate the AL 288-1 models (n = 1, specimen ID: Subject03) (Charles et al., 2020). Subject03 was an adult female aged 26, weighing 72.6 kg and 176 cm in height. The Wiseman (Wiseman, 2023) Australopithecus afarensis model was used, based upon the AL 288-1 specimen, with the foot belonging to specimen AL 333-115 (Desilva et al., 2018); see further details in Supplementary Information 1. Both models include 36 muscles of the right side of the pelvis and right lower limb, crossing the hip, knee, ankle and metatarsophalangeal joints (Table 1). Both models each had 16 degrees of freedom (DOF): three rotational DOFs in the pelvis (pitch, tilt and roll), three rotational DOFs in the right hip (flexion-extension, adduction-abduction and long-axis rotation) and one DOF each in the right knee, ankle and metatarsophalangeal joints (flexion-extension). All other DOFs (i.e., in the left limb and translational) were fixed (locked).

Muscular parameter estimation

Seven Au. afarensis (AL 288-1) models were generated, each using different estimation methods for muscle parameters, alongside one human model that was subject-specific. These estimations for AL 288-1 produced either a model composed of muscles that ignored tendon compliance and muscle force-length-velocity relationships (referred to as the ‘static-muscle’ approach), or a model composed of ‘fully’ Hill-type muscles which included an elastic tendon and used force-length-velocity relationships (i.e., the ‘dynamic’ approach; see: Zajac, 1989; Millard et al., 2013). All muscles in the human model followed the dynamic approach. Four different methods were used to estimate architectural properties of the muscles in AL 288-1:

1. Alpha shape centroid estimation (method assumes an isometric ‘static-muscle’ model with no muscle pennation and was thus not suitable for use in the ‘dynamic-muscle’ model which requires pennation inputs)

2. Convex hull centroid estimation (static- and dynamic-muscle models)

3. Arithmetic centroid estimation (static- and dynamic-muscle models)

4. 3D model-based estimation (static- and dynamic-muscle models)

Each variant was based upon a dataset (n = 22) composed of published muscle architecture data from the human lower limb (Charles et al., 2020; Friederich & Brand, 1990; Charles, Moon & Anderst, 2019), which included muscle belly mass (m)muscle), optimal fibre lengths (ℓo) and optimal pennation angle (αo). This approach follows the estimation methodology developed by Bishop, Cuff & Hutchinson (2021a), but was modified here to include the 3D muscle model approach (Demuth, Wiseman & Hutchinson, 2023b) in which the polygonal muscles reconstructed for AL 288-1 (Wiseman, 2023) provided the muscle mass values for the subsequent calculation of the physiological cross-sectional areas and isometric force (Fmax) in the 3D model-based variants. The approach was further modified to include αo estimates (code provided in Supplementary Information 2). These approaches calculate the normalised muscle mass (m∗), normalised fibre length (ℓ∗) and αo (non-normalised). The dynamic-muscle models (mDynamicMuscleModel∗) defined m∗as: (1) mDynamicMuscleModel∗=mmusclembody

and for the static-muscle models (mStaticMuscleModel∗), was defined as: (2) mStaticMuscleModel∗=mmuscle⋅ cosαombody

in which mbody was the individual’s body mass. ℓ∗ was defined in both approaches as: (3) ℓ∗=ℓombody0.33.

Three values were extracted for each muscle from the dataset (see example in Fig. 1): (1) the average of the input values provided the arithmetic value (i.e., the average of all data points), (2) a 3D convex hull was applied to the data points and the centre point of the hull provided the convex hull value calculated by weighting the centroid of each triangle by the respective face area in comparison to the total surface area of the convex hull—this was mathematically defined by Demuth et al. (2022), and (3) the largest circle which can be fit within the 2D convex hull produced an alpha shape of which the centre point provided the alpha value (Fig. 1). The arithmetic values were used as the input data for the 3D model variant, in which parameters were identical to the arithmetic approach, but the resultant PCSA and Fmax calculations were based upon the polygonal muscle mass calculations. Whilst αo was incorporated into the above equations for the static-muscle model, the value for αo was left at 0° in OpenSim.

Table 1 Muscles included in this study and their abbreviations.

Muscles are loosely ordered proximally-distally.

Abbreviation	Muscle name	Abbreviation	Muscle name	
AB	Adductor brevis	BFL	Biceps femoris (long head)	
AL	Adductor longus	BFS	Biceps femoris (short head)	
AM	Adductor magnus	VI	Vastus intermedius	
GemInf	Gemellus inferior	VL	Vastus lateralis	
GemSup	Gemellus superior	VM	Vastus medialis	
ObtExt	Obturator externus	MG	Medial gastrocnemius	
QF	Quadratus femoris	LG	Lateral gastrocnemius	
TFL	Tensor fasciae latae	PB	Peroneus brevis	
GMax	Gluteus maximus	PL	Peroneus longus	
GRA	Gracilis	SOL	Soleus	
GMed	Gluteus medius	TA	Tibialis anterior	
GMin	Gluteus minimus	TP	Tibialis posterior	
ILI	Iliacus	PECT	Pectineus	
PIRI	Piriformis	POP	Popliteus	
RF	Rectus femoris	EHL	Extensor hallucis longus	
SAR	Sartorius	FHL	Flexor hallucis longus	
SM	Semimembranosus	EDL_DIGITS I-IV	Extensor digitorum longus	
ST	Semitendinosus	FDL_DIGITS I-IV	Flexor digitorum longus	

In total, our workflow produced seven different estimates of architecture per muscle (n = 36) in the AL 288-1 pelvis and lower limb. There were four static-muscle models based upon (1) an alpha shape, (2) a 2D convex hull, (3) the arithmetic centre, and (4) a 3D model; and three dynamic-muscle models based upon (1) a 3D convex hull, (2) the arithmetic centre, and (3) a 3D model. See Fig. 1 for an example of how different these parameters are between variants in one muscle example (data available: Wiseman, Charles & Hutchinson, 2024).

Each muscle’s physiological cross-sectional area (PCSA) for all seven models was calculated as: (4) PCSA=cosαo⋅mmuscleρ⋅ℓo

where p is tissue density of 1,060 kg m−3 (Mendez & Keys, 1960), for the 3D muscle models, volumes were equivalent to mmusclep−1. αo was ignored in the dynamic-muscle model’s Eq. (4) because it is separately accounted for in the OpenSim pipeline (Seth et al., 2011; Millard et al., 2013; Cox et al., 2019). Each muscle’s maximal isometric force (Fmax) subsequently was calculated as: (5) Fmax=PCSA⋅σ

Figure 1 Examples of muscle architecture estimation.

The M. soleus muscle is shown here as an example. (A) The static-muscle model, in which an alpha shape was fit to the plotted data points of modern human values. An alpha centroid, an arithmetic centroid and convex hull centroid were all calculated, thus providing three estimates per muscles. A 3D-model based approach was also included using the arithmetic outputs but with muscle mass values provided from the 3D muscle reconstructions (Wiseman, 2023). (B) The dynamic-muscle model, in which pennation angle was included in architectural estimations producing a 3D alpha shape to provide an arithmetic centroid and convex hull centroid. A 3D- model-based approach was also included. The projected static-muscle model approach is shown for reference. Values were normalised and scaled to be between 0 and 1 so as to remove the potential effects of different magnitudes of the axes. The ‘input parameters’ refer to the values directly input into the OpenSim model.

where σ is the muscle’s maximal isometric stress, with a representative value of 300 kN m−2 used across all muscles (Medler, 2002; Wells, 1965; James, Altringham & Goldspink, 1995; Zajac, 1989; Maganaris et al., 2001; Saito et al., 2021; also Hutchinson, 2004a; Hutchinson, 2004b and references therein). The specific stress values used do not matter for our simulations, which do not attempt to estimate true in vivo maximal performance, but rather focus on data exploration. Tendon slack length (ℓS) was calculated for the dynamic-muscle model variants only, with the assumption that muscle fibres had the capability to range in length from 0.5 to 1.5 × ℓo throughout the muscle–tendon unit’s length across the pertinent limb joints’ ranges of motion (Manal & Buchanan, 2004). For the static-muscle models, tendon slack length was left at a default value of 1 m. All muscles were modelled in OpenSim using the Millard muscle class (Millard et al., 2013). For the dynamic-muscle models only, force-length-velocity properties were included through the Static Optimisation procedure. For the static-muscle models, this relationship was ignored.

For the human, muscle architecture from ‘Subject03’ was previously determined via diffusion tensor imaging and muscle paths and masses were established via muscle resonance imaging (Charles et al., 2020), producing a subject-specific model rather than a generic model. The human model is provided here as a comparative model to the australopith, rather than as an additional model for examining the influences of static versus dynamic muscles or different architecture estimates. The impacts of variations in muscle force-generating properties on inferences of muscle function in humans have been previously described (Charles et al., 2022; Kramer et al., 2022).

Inverse simulations

A previously captured walking cycle was used (Wiseman et al., 2022), tracked and scaled to the musculoskeletal model of the human model using the ‘model scale’ and ‘inverse kinematics’ tools in OpenSim 4.3 (Delp et al., 2007; Seth et al., 2018). In this previous study, one participant walked with a typical walking posture (five trials along a 12 m long trackway) at a speed of 1.0 m/s and a 14-camera optoelectronic 3D motion capture system (250 Hz, Oqus Cameras, Qualysis AB, Gothenburg, Sweden) was used to capture kinematics via a skin-attached reflective marker-set. A mid-stance (30% of the gait cycle), posture was extracted from each right limb’s stride, averaged and applied to the human model in OpenSim (Supplementary Information 3). The left limb was modelled in a mid-swing pose at 30% of the gait cycle, with toe clearance ensured. Each left limb segment had inertial properties, but all DOFs were locked.

These joint angles were used as a guideline to position the AL 288-1 model in a similar mid-stance pose. However, it was not possible to directly apply the joint angles from the human to AL 288-1 due to geometric differences (e.g., Gatesy & Pollard, 2011) which resulted in the foot mis-aligning with the ground (i.e., a flat foot on the ground was a requirement of this study). This possibly was due to differences in iliac flaring of the pelvis and lower limb proportions (Lovejoy, 2007; Lovejoy, 2005; Jungers, 1982; Stern & Susman, 1983; Ward, 2002; Wiseman, 2023) influencing foot placement. Slight manual modifications were made to the hip joint angle to position the limb with the foot flat on the ground (i.e., greater hip abduction and knee adduction were required; knee adduction angle was subsequently locked). These joint angles were applied to all seven AL 288-1 models.

To maintain dynamic congruity between the model and the experimentally generated ground reaction forces (GRFs) and kinematic data (e.g., the hypothetical positioning of AL 288-1 into a mid-stance single-limb support pose) during static optimization (i.e., consistency between the simulated model’s GRFs and kinematic data), residual actuators—i.e., additional forces that help to achieve dynamic congruity by adjusting the model to match the input data more closely during static optimisation—were applied (Mx,y,z and Fx,y,z, in which the x-axis is anterior-posterior, the y-axis is perpendicular to the ground, and the z-axis is medio-lateral) to the pelvis-body COM (see: Hicks et al., 2015). The activations of these actuators during the simulation are heavily penalised by the optimisation algorithm and were thus only recruited if necessary. Results from these residual actuators as outputs from the simulations can be found in Supplementary Information 4, showing that they were well below the tolerances recommended by Hicks et al. (2015).

Following Bishop, Cuff & Hutchinson (2021a), gravity was increased accordingly relative to mbody during the static simulations to produce net force balance and to help achieve dynamic consistency (see Fig. 2). For example, in the australopith, the vertical GRF (vGRF; see below) applied at 1.6 * BW was 544.171 N and the corresponding gravitational force was changed to 15.691 m/s2. By increasing gravity relative to mbody, the gravitational force acting on the body is artificially enhanced. This increase in gravitational force helps achieve balance with the applied vGRF (Bishop, Cuff & Hutchinson, 2021a). This ensured that there was force equilibrium (i.e., all forces were balanced, assuring no net acceleration) when the applied vGRF was smaller or greater than 1 * BW.

Figure 2 Human and AL 288-1 musculoskeletal models with each body segment’s COM (small green balls) and the specimen’s COM (large green ball) shown, alongside each joint’s coordinate system.

Both specimens are positioned in a midstance, erect posture with single limb support. A vGRF was then applied to the foot (green arrow in image on the right).

Vertical and medio-lateral GRFs were applied to the static inverse simulations (all joints had zero accelerations and velocities). No antero-posterior GRFs were appended. The centre of pressure of the foot (point of GRF origin) was assumed to be at the midpoint between the COMs of the pes (defined here cumulatively as the hindfoot, midfoot and metatarsals) and digit segments. The medio-lateral GRF vectors were calculated as the ratio between the vertical and mediolateral vectors from a large dataset (GAITREC) of published GRF profiles (n = 211 healthy participants; n = 2,382 trials in total) extracted at 30% of the gait cycle at a controlled walking speed of 0.98 m/s (Horsak et al., 2020b; Horsak et al., 2020a). Because the data were not evenly distributed, the median ratio of 0.011265 was used rather than the mean. The mediolateral GRFs were thus appended in AL 288-1 as a fraction of the vGRF. The vGRFs were input as a factor of body weight (BW) starting at 0.2 * BW and increased by increments of 0.2 * BW (i.e., see Supplementary Information 1) until the static optimization algorithm used for the inverse simulations could no longer find a solution to achieve static equilibrium for any combination of muscle activations (i.e., simulation failure), while minimising the sum of squared activations for each DOF (n = 16) (Rowninshield & Brand, 1981).

Evaluation of simulations

Simulated muscle activations from the human were qualitatively compared to published electromyography (EMG) studies that report on walking in healthy, non-pathological individuals (Cappellini et al., 2006; Van Criekinge et al., 2018) to evaluate the reliability of the human simulation (Hicks et al., 2015). We inspected the mid-stance on-off timings of the EMG signals, but ignored magnitudes because this study only modelled a static, mid-stance pose (See: Supplementary Information 5). Subsequently, we set the performance criteria of the AL 288-1 simulations (e.g., Demuth, Wiseman & Hutchinson, 2023b) as follows: First, we used the maximal amount of vGRF exertion in the human as a benchmark for the expected vGRF exertion (vs. BW) in the AL 288-1 simulations, assuming that two phylogenetically closely related and morphologically similar individuals employing similar locomotor behaviours should be able to sustain comparable forces on an extended limb. If a simulation was unable to support the body below this benchmark, the underlying model was assumed to be ‘weak’ in light of the current setup in which a hypothetical limb pose was modelled to examine the range of outputs that can be generated if muscle architectural parameters are changed. A different limb posture would likely generate different results in both the human and AL 288-1 individuals and would lead to the selection of a different benchmark value.

Second, the operating fibre lengths, muscle specialisation (quantified as the ratio of ℓo to PCSA) and simulated activations of the AL 288-1 models were compared to those of the human model and simulations to provide preliminary insights into functional differences between species.

Results

How do the different estimation methods for muscle parameters influence maximal performance (sustainable ground reaction force)?

The human model was able to exert a maximum of 2.2 * BW vGRF on a single limb at 30% of the gait cycle. The maximally exerted vGRF in the AL 288-1 simulations differed according to model variant (Table 2). The static-muscle models typically weakly supported the capability of this representative mid-stance pose compared with the human results as a baseline, with some AL 288-1 variants unable to withstand >1.8 * BW vGRF on a single limb. Contrastingly, the static-muscle model 3D variant withstood 3.6 * BW vGRF, indicating that the muscle mass estimations following the alpha and 2D convex hull approaches possibly were underestimated and thus under-performed. However, the 3D muscle modelling approach produced muscles with greater force capacities and greater sustained vGRFs. The dynamic-muscle models were able to withstand greater vGRF than the static-muscle models, excluding the 3D variant. All dynamic-muscle models indicated that the limb was capable of supporting greater vGRF than the human, ranging between 2.6 * BW (the convex hull variant) to 2.8 * BW (the 3D variant) vGRF. Notably, the dynamic-muscle 3D variant was markedly weaker than the static-muscle 3D variant. Overall, the average amount of vGRF that the limb could withstand in the static-muscle models was 2.2 * BW and in the dynamic-muscle models was 2.6 * BW. Only the static-muscle models (excluding the 3D variant) were below the 2.2 * BW benchmark and are thus assumed to be ‘weak’.

Table 2 The maximum GRF that each model variant can sustain, in which the maximum GRFs were applied to the model as a factor of body weight (BW) of the individual in the simulations.

For example, a value of 2.2 indicates that the model can sustain 2.2 * BW.

Model	Muscle model type and variant	Maximum possible GRF as factor of BW	Maximum GRF (N)	
Human	Subject-specific	2.2	1443	
AL 288-1	Static-muscle–Alpha variant	1.6	612	
Static-muscle–Convex Hull variant	1.6	612	
Static-muscle–Arithmetic variant	1.8	680	
Static-muscle–3D model variant	3.6	1088	
Dynamic-muscle–Convex Hull variant	2.6	1156	
Dynamic-muscle–Arithmetic variant	2.8	952	
Dynamic-muscle–3D model variant	2.8	884	

Do the different estimation methods for muscle parameters produce muscles with different inferred architectural specialisations (force versus velocity?)

We used the ratio of ℓo to the PCSA to investigate how the different variants influenced whether a muscle was specialised more for force (∼small ratio value) or for velocity (∼high ratio value) (Fig. 2). Whilst is impossible to dictate a ‘cut-off’ value to distinguish between low versus high ratios, the general trend across muscles can instead be examined to elucidate differences (i.e., Charles et al., 2022). Small changes in ratio are expected because architectural parameters are not identical between each estimation method or variant, but larger changes between variants indicate major functional changes of a muscle—for example, if a variant’s ratio is twice as large, then we can interpret this as a difference in predicted muscle specialisation. Generally, most muscles across the variant spectrum appeared similarly specialised (small changes such as those found in the GRA, GMed, SM, ST, EDL, PL, PB, and SOL muscles, for example, were negligible). Some muscle specialisations switched depending on the variant used. For example, the static-muscle model’s variant estimates for the MG (Fig. 2D) predicted that this muscle was velocity-specialised (∼higher ratios), but the dynamic-muscle model’s variants for the same muscle instead predicted that the muscle was suited for force-generation due to lower ratios that were 3.2× smaller than the ratios of the static-muscle variants. The static-muscle 3D-based model of the TP (Fig. 2E) predicted a velocity-specialised muscle, yet all other variants predicted a force-specialised TP (i.e., the 3D-based variant was almost 3× the amount of the other ratios, a profound difference). In the VL (Fig. 2C), the dynamic-muscle variants suggested that this muscle was a velocity-specialised muscle, but the ratios of the static-muscle model variants were 2.2× greater; more force-specialised. In the GMax, both static-muscle and dynamic-muscle models of the 3D variant were found to have higher ratios (velocity-specialised) than all other variants. Overall, there was broad similarity among variants with only a few aforementioned outlying differences.

Do any of the methods produce a muscle which could generate relatively greater forces or operate on a different part of its force-length curve?

We sought to determine if any of the variants produced a muscle which generated relatively greater forces. Using the plot function in OpenSim, we plotted each muscle’s ℓ∗ from the three dynamic-muscle models of AL 288-1 and also for the human for comparison of ℓ∗ only (Fig. 3). In the AL 288-1 models, the convex hull and arithmetic variants had comparably lower muscle force (maximum: ∼330 N and ∼220 N, respectively; Figs. 3C, 3D) than that of the 3D variant (∼580 N; Fig. 3B).

Figure 3 Boxplot of the normalised fibre length of each muscle in the human and the three dynamic-muscle model variants in the australopith.

A blue colour indicates that the fibre is operating on the descending limb of its force-length curve, black is the plateau, and red indicates that the fibre is operating on its ascending limb.

Next, we assessed if a muscle’s force-length curve was the same or different between each of the variants based on their normalised fibre length values from the chosen single time point of a walking stride A value >1 was on the descending limb, and a <1 was on the ascending limb. The patterns of where fibre lengths lay on their force-length curves were broadly similar between the convex-hull and arithmetic variants, with some differences in the 3D model-based variant. For example, the AB operated on the same region of its force-length curve in all variants, but had a muscle force of around 10 N in the 3D model-based variant (Fig. 3B) in comparison to greater force in the other variants, ranging between 75–250 N (Figs. 3C, 3D). The AM was on the descending limb (i.e., actively lengthened) for the 3D model-based variant (Fig. 3B), but around the plateau of the curve (near-isometric) for the other variants (Figs. 3C, 3D). The AL was at the plateau of the curve in the 3D model-based variant (Fig. 3B), but instead was on the descending limb in the other variants (Figs. 3C, 3D).

How do these different methods affect simulated influence muscle activation patterns (i.e., different groups activated, or the same muscle to different activation levels)?

Simulated muscle activations during single-limb stance in the AL 288-1 models are reported in Fig. 4. Muscles are heuristically simplified into primary functional groups. The static-muscle models (excluding the 3D variant) were broadly comparable with heavily activated ankle plantarflexors (LG, PB and SOL) and moderately activated hip adductors, hip extensors and knee extensors. In the 3D static-muscle variant (Fig. 4A), there was increased activation in comparison to the other static-muscle variants (Figs. 4B–4D) in the hip adductors and knee extensors. The 3D static-muscle variant also had different patterns in the plantarflexors in which maximal activation only occurred at maximal vGRF (i.e., just before simulation failure), rather than being activated earlier in the simulations at ∼0.8 * BW vGRF as per the other static-muscle variants.

Figure 4 Simulated muscle activations for the australopith models (seven variants) and the human model.

The static-muscle models are on the top row, the dynamic-muscle models are the bottom row.

The muscle activations in the dynamic-muscle variants were mostly similar to the static-muscle models with few differences. In the dynamic-muscle convex (Fig. 4G) and 3D variants (Fig. 4E), the TP and FHL muscles were maximally activated, but had low activation in the dynamic-muscle arithmetic variant (Fig. 4F). The dynamic-muscle arithmetic variant (Fig. 4F) was the only variant to not have maximal activation of the PB. Activations patterns overall were comparable. The activation patterns of the SM and ST were switched between the dynamic-muscle convex (Fig. 4G) and arithmetic variants (Fig. 4F), in which the SM had greater activation in the arithmetic variant, but in the convex hull the ST muscle activation was greater, and the convex and arithmetic variants also were the only ones to have highly activated hip adductors. Both 3D variants were the only variants to not activate the SM and ST muscles. The dynamic-muscle 3D variant (Fig. 4E) had reduced hip extensor activation in comparison to the other dynamic-muscle variants.

In Fig. 5, the maximum activation of each muscle (i.e., from maximal sustainable vGRF) from each of the variants is presented, alongside the average activation and standard deviation across all variants. Whilst activations of most muscles across each of the variants are broadly comparable, such as the SOL which was fully activated (full activation = 1) across all variants, other muscles had activations that depended more on the variant. The TP was highly activated in the dynamic-muscle 3D and convex hull variants, but weakly activated in all other variants at the point of maximal vGRF exertion. The PL, TA and PB had similar patterns, with activations that were variant-dependent.

Figure 5 Maximum simulated muscle activations for each of australopith variants, which were then averaged and presented with the standard deviation.

Whilst some muscles have similar patterns of activation (e.g., the AB), other muscle activations are dependent upon the variant (e.g., the TP). ‘Arithm’ = arithmetic; ‘ConvHu’ = convex hull; ‘Static’ = static-muscle model; and ‘Dynamic’ = dynamic-muscle model.

Next, we calculated the mean activations in each major muscle group (see Supplementary Information 6). All variants mostly produced comparable mean activations per muscle group with only minor deviations, such as increased activations of hip extensors in the dynamic-muscle arithmetic variant in comparison to the other variants. Only the 3D (static- and dynamic-muscles) and dynamic-muscle convex hull variants had a minimally activated ankle dorsiflexor group, which was inactive (activation = 0) across all other variants.

What are the differences in muscle activations and force length states between species?

The simulated muscle activations in the human model were consistent with published EMG studies of muscle excitation at mid-stance in healthy individuals (Perry & Burnfield, 2010; Van Criekinge et al., 2018; Cappellini et al., 2006; Wall-Scheffler et al., 2010) and are reported in Supplementary Information 5. As such, the human simulation was considered sufficiently representative.

In both the human and the AL 288-1 simulations, it was the ankle plantarflexor group that limited maximal vGRF exertion (Fig. 4). The hip adductors were only activated in the australopith, not in the human, alongside more moderate activations in the hip extensor and knee extensor groups. The human was the only model in which the PIRI muscle was fully activated. Overall, more muscles in the AL 288-1 limb were required to maintain single-limb stance than in the human, but these muscles were only mildly to moderately activated.

Similar patterns in the simulated operation of ℓ∗ on its force-length curve were observed between the human and AL 288-1 variants for the mid-stance pose (Fig. 3). For example, the MG in all variants generated high force in comparison to the other muscles and was on the descending limb of its force-length curve. The FDL and EDL were the same (on the descending limb) in all AL 288-1 variants, but different in comparison to the human, in which the FDL and EDL instead were on the ascending limb (actively shortened) or were at the plateau of the curve (isometric) at 30% of the gait cycle. The vastus muscle group was on the ascending limb in the human, but around the plateau of the curve for all AL 288-1 variants in which this muscle group was weakly activated. The SM was on the ascending limb for the human (but inactive) whereas it on the descending limb for all AL 288-1 variants. The SOL was activated in all AL 288-1 variants, but only weakly activated in the human model (descending limb), and heavily activated in the AL 288-1 variants (force-length plateau).

Discussion and Conclusion

Here, seven different Au. afarensis musculoskeletal models were created, all identical in form and creation apart from the input architectural properties that were estimated in different ways. Static (inverse) simulations were computed on a single limb for each model variant which was representative of a mid-stance posture during typical walking in a modern human. We explored the diverse outcomes resulting from these different modelling choices to elucidate how these choices could impact simulation results, which is important for choosing how to model extinct taxa. Such an approach should be considered for future studies that create such models: do outputs hold up in a specimen that cannot be empirically tested if the underlying muscle architecture is changed? Fortunately, broad comparability in the simulated outputs is promising, with only a few caveats which are discussed below. Future studies should consider the likelihood of different simulation outputs if model conditions are altered. For instance, if a researcher creates a model and simulates forward walking in a hominin and observes maximal recruitment of all muscles and thus high metabolic costs, it is essential to assess if/how altering the underlying muscle parameters might enhance the model’s performance, and vice versa.

Because our study only adopted a static (inverse) approach, the results do not equate to what the limb will realistically do during the mid-stance phase of the gait cycle (Cappellini et al., 2006), but rather how the muscles support single-limb stance in a model that as zero accelerations and velocities, with no antero-posterior GRFs appended. Nevertheless, the muscles in the human model overall matched well with published EMG data sampled during human locomotion (i.e., Perry & Burnfield, 2010; Van Criekinge et al., 2018; Cappellini et al., 2006; Wall-Scheffler et al., 2010). In light of the requirements of the current study, we contend that the human model is useful and replicable of human muscle activity, especially as our study did not attempt to infer true in vivo maximal vGRF exertion. In particular, the adopted pose in AL 288-1 was an informed hypothetical limb position based upon an empirically-tested human model used to address our study’s main aims, which are more methodologically oriented.

Our first question asked if the different estimation methods for muscle parameters influenced maximal performance (sustainable vGRF) in the simplified scenarios. Variants of the same approach (i.e., all dynamic-muscle variants versus all static-muscle variants) were broadly comparable, with just a few differences, notably the maximal vGRF before limb collapse. The static-muscle variants produced weaker muscles, and struggled to maintain single limb support at >2 * BW vGRF, although the 3D model-based variant was stronger (see below). The static-muscle model approach using 3D volumetric muscle reconstructions might be suitable for studies which do not require specific muscle activation patterns or complex results. Importantly, and perhaps surprisingly, the 3D variant with a static-muscle model outperformed that with a dynamic-muscle model (3.6 versus 2.8 BW * vGRF). We acknowledge that the joint angles were slightly disparate between the human and AL 288-1 to achieve a flat foot on the ground and, consequently, values lower than the 2.2 * BW vGRF benchmark may not necessarily mean underperforming models. Nevertheless, in light of the goals of the study which include data exploration to ascertain the sensitivity of a model to changing input parameters, the results of the study are upheld.

Next, we asked if these different methods produce muscles with different inferred architectural specialisations (force versus velocity)? We demonstrate how each of the variants influenced muscle specialisation. Muscle specialisation is determined by each muscle’s ℓo and PCSA, whether for force (shorter fibres; low ratios of ℓo versus PCSA) or velocity (long fibres; high ratios) (e.g., Charles et al., 2020; Payne et al., 2006; Sharir, Milgram & Shahar, 2006). Here, we used the ratio of ℓo:PCSA to establish how muscle specialisation can be influenced by each of the estimation methods. Whilst ℓo values were the same for different variants of the dynamic-muscle and static-muscle models (i.e., both convex-hull variants have the same ℓo), the PCSA values were different owing to the inclusion/exclusion of αo (see: Eqs. (2) and (4)). Our results demonstrate that the exclusion of an elastic tendon (i.e., incorporation of the tendon force-length relationship) and αo (the static-muscle model approach) produced models that were weaker and required greater muscle activation for single-limb stance (not including the 3D variant). In contrast, the dynamic-muscle models typically were stronger, sometimes with muscle specialisation that differed from that of the static-muscle models. For example, one variant (the dynamic-muscle model: arithmetic variant) produced a force-specialised MG muscle, but another variant (the static-muscle model: arithmetic variant) had the corresponding muscle as velocity-specialised instead. We also estimated that the muscles from each of the variants operated on the same parts of their force length curves, with only a few differences observed in the adductors of the 3D model-based variant. We further estimated that the 3D variants had greater maximal muscle forces than those of the convex and arithmetic variants.

We also sought to understand if each of the methods affected simulated muscle activation patterns (i.e., were different groups activated, or the same muscle to different activation levels?). Activation patterns were mostly similar between the variants, with only a few differences observed in the 3D variants (dynamic-muscle and static-muscle) versus the other variants, although most differences related to magnitude rather than differences in specific muscle activation.

Variant selection in a future study will depend on the outcomes sought and the design of the study in question alongside the required level of modelling complexity (i.e., Demuth et al., 2023a). If a study seeks to model dynamic movement with simulated muscle activations, then we would advocate that a dynamic-muscle model with an elastic tendon is best. The static-muscle model variants are weaker (except for the 3D muscle variant) and might under-perform in studies employing forward dynamics of gait. However, the static-muscle model approach (convex-hull and arithmetic variants) would be acceptable to use in future studies that do not simulate activities requiring greater GRF exertion, such as running (e.g., Keller et al., 1996) or jumping (e.g., Simpson et al., 2018), assuming that limitations of the approaches are fully acknowledged. There are, however, several scenarios in which a researcher may choose to incorporate a static-muscle model. For example, Demuth, Wiseman & Hutchinson (2023b) opted to use a static-muscle model in a static simulation to test the limb posture of an extinct taxon whereby no dynamic motions were modelled, and the addition of an elastic tendon would have added unnecessary complexity to the model. Furthermore, dynamic simulations can be computationally intensive and the addition of an elastic tendon in an already complex musculoskeletal model and simulation would exponentially increase computational time and effort (i.e., Bishop et al., 2021b), which may not be necessary in light of the research question (see Demuth et al., 2023a). Assuming that a researcher has investigated the sensitivity of elasticity versus non-elasticity in a simple simulation (i.e., by using the static optimisation tool in OpenSim Delp et al., 2007), the researcher can then opt for static-muscle models to ensure faster convergence in their simulations, or ask simpler questions of those simulations, as demonstrated by Bishop et al. (2021b) in their work assessing the dynamic role of the tail during locomotion in the extinct dinosaur Coelophysis.

We next asked: is there a preferential muscle modelling variant to use? First, we consider the static-muscle model approach. Variants were mostly comparable, but the 3D model-based variant was stronger in the static-muscle model approach, with muscles that had greater force. The 3D model-based variant requires muscles to be reconstructed that are the best-informed estimate; yet, this is more subjective if the geometric differences between the extant and extinct species are profound (Demuth et al., 2022). If this variant is to be used, then the researcher should ensure that the geometric differences are minor, otherwise they might risk over- or under-estimating muscle volumes. The convex hull and/or the arithmetic variants are recommended if an appropriate comparative living specimen cannot be used to guide 3D muscular recreation in extinct taxa, although limitations should be duly noted.

Variant choice in the dynamic-muscle model approach is less obvious, whereby all variants produced comparable results and architectural properties did not appreciably alter interpretations. Therefore, we do not recommend one variant over another, but we advise that different variants might slightly influence muscle specialisations and activations.

By assuming that musculature is highly evolutionarily constrained (i.e., Wiseman, 2023), the human model was used here as a benchmark to evaluate the AL 288-1 model. We emphasise that only muscle parameters measured from a human were used to infer the parameters of the AL 288-1 model, yet a phylogenetically-bracketed model should ideally be informed by measured architectural parameters from other species, such as a chimpanzee (see: O’Neill, Nagano & Umberger, 2023). Nevertheless, this study focused on data exploration to determine the sensitivity of simulation outputs when the underlying architectural parameters were changed. Future studies may wish to re-calculate these parameters by including measured architectural parameters from a greater range of species. We also acknowledge that the selection of a different maximal isometric stress value would change the point in which the static optimisation algorithm would fail. For example, a higher stress value in the human would create a model that could withstand greater than 2.2 * BW vGRF.

There are also limitations to using Hill-type muscle models in both the static-muscle and dynamic-muscle approaches. Despite the Hill-type muscle model being a widely used model across biomechanics (e.g., Seth et al., 2018), there are known limitations regarding the accuracy of the muscle model in representing the intricate dynamics of muscle contraction (e.g., assumes homogenous muscle properties across muscle lengths, does not consider titin filament dynamics, etc; see: Zajac, 1989; Millard et al., 2013).

In conclusion, different ways to estimate muscle architectural properties will influence a muscle’s specialisation and activation, and more-so can produce a model which under-performs and is too weak. A researcher can create two identical musculoskeletal models of an extinct taxon and by only changing the input architectural parameters and including/excluding an elastic tendon, can have one version of the model that is weak and unable to support the body posed on an extended limb and another version that can support the body in multiples of body weight on a single limb which could be inferred as the capability to run and jump. We demonstrate here the range of outputs that are generated only by changing the underlying parameters. These findings carry significant implications for the creation and simulation of extinct taxa, especially when empirical tests are challenging or impossible. Acknowledging that altering underlying assumptions can yield varying outputs is important for such simulation studies. Promisingly, we found broad replicability within each of the estimation methods (static versus dynamic) tested here. The only major disparity of concern is model strength, which itself might not be a concern for some research aims. Therefore, the choice of parameter estimation method should be carefully selected alongside the requirements of the study. A dynamic-muscle model with an elastic tendon (Zajac, 1989) probably is the ideal approach for future studies that simulate locomotion with musculoskeletal models of extinct species. However, we have shown that there is some value in ‘static-muscle’ models for answering more basic questions about locomotion; especially if muscle volumetric models are used for estimating input parameters.

Supplemental Information

Supplemental Information 1 Musculoskeletal model creation

Click here for additional data file.

Supplemental Information 2 MATLAB code for estimating muscle parameters

Code tested on MATLAB versions 2021 onwards.

Click here for additional data file.

Supplemental Information 3 Joint angles (degrees) used in each of the quasi-static inverse simulations

Click here for additional data file.

Supplemental Information 4 ‘Residual actuators’ at the pelvis-body segment in the human and AL 288-1 simulations

Click here for additional data file.

Supplemental Information 5 Comparison to EMG studies

Muscle activations in the human musculoskeletal model in this study are reported at 30% of the gait cycle in comparison to published electromyography (EMG) studies. Here, simulated muscle activations as per the static optimisation outputs are used. No muscle activation dynamics have been modelled and, in this approach, simulated excitations are not produced.

Click here for additional data file.

Supplemental Information 6 Mean activations per group

Simulated grouped mean muscle activations for the australopith models (seven variants).

Click here for additional data file.

We thank Ajay Seth, Oliver Demuth, Andrew Cuff and Julia van Beesel for their insightful conversations. We also thank three anonymous reviewers and the editor for their helpful feedback.

Additional Information and Declarations

Competing Interests

Author Contributions

Data Availability

John R. Hutchinson is an Academic Editor for PeerJ.

Ashleigh L.A. Wiseman conceived and designed the experiments, performed the experiments, analyzed the data, prepared figures and/or tables, authored or reviewed drafts of the article, and approved the final draft.

James P. Charles performed the experiments, authored or reviewed drafts of the article, and approved the final draft.

John R. Hutchinson performed the experiments, analyzed the data, authored or reviewed drafts of the article, and approved the final draft.

The following information was supplied regarding data availability:

The data is available at the University of Cambridge Repository: Wiseman ALA, Charles P. & Hutchinson J.R (2024). Dataset for: Static versus dynamic muscle modelling in extinct species: A biomechanical case study of the Australopithecus afarensis pelvis and lower extremity. Apollo - University of Cambridge Repository. https://doi.org/10.17863/CAM.96503.

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
