# Peer review of "Static versus dynamic muscle modelling in extinct species: a biomechanical case study of the Australopithecus afarensis pelvis and lower extremity"

_PeerJ, doi:10.7717/peerj.16821_

## Round 0.1 · original submission · Major Revisions

Thank you for submitting your manuscript, “Simple versus complex muscle modelling in extinct species: A biomechanical case study of the Australopithecus afarensis pelvis and lower extremity” for consideration in PeerJ. We have received evaluations from three external reviewers. Though Reviewer 1 recommends Rejection, based on the publication status of the Australopithecus musculoskeletal model that forms the basis for this study, I’m in agreement with Reviewers 2 and 3 that the information provided in the submitted manuscript is sufficient to form an evaluation.

Both Reviewers 2 and 3 cite the need for clarifying the motivation behind the research, the specific analytical approach, and the interpretation of results. I agree with both reviewers that the requested changes are of sufficient magnitude to fall under “major revision”. I urge the authors to take a detailed look at their comments and suggestions.

Additionally, I have a few small editorial suggestions:

(line 73) Missing a period after “isometric force”.

(line 118 and line 341) “Bishop and colleagues (2021)” – the references list three papers that Bishop published with colleagues in 2021. Which is this?

(lines 271-273) The listing of morphometric parameters of Subject 03 is already presented lines 177-178. This statement is thus redundant.

(lines 320-323) This statement makes no sense to me. Specifically, “prevent from becoming fully activated if they are sufficiently large enough to never be fully activated”. Does this just mean that reserve torques had sufficient magnitudes to cover the required range?

(lines 336-338) What do the X, Y, and Z axes represent in the global coordinate frame?

(Figure 1) I could not discern the difference between the blue and black crosses in Figure 1b. Perhaps different symbol could be used, in addition to colors (i.e., one could be an “X”).

Reviewer 1 ·

Basic reporting

Insufficient background and literature referenced.

Experimental design

The manuscript here fails significantly. The entire paper rests on an australopithecine model described in Wiseman (2022). This citation, however, is not a peer-reviewed paper but instead only a preprint on BioRxiv. This is entirely inappropriate. It fails on meeting both technical and ethical standards. This manuscript was submitted prematurely and should not receive full review.

Validity of the findings

It is not possible to evaluate the validity of the finding because the model which provides the foundation for this paper has not been reviewed and only exists as a preprint.

Reviewer 2 ·

Basic reporting

In this article Wiseman et al. investigate how different muscle parameters affect various aspects of muscle performance in a musculoskeletal model of Au. afarensis. The writing and language are clear throughout, and the authors provide good background on the considerations concerning the application of muscle models and muscle architecture to musculoskeletal model simulations.

A minor issue with the introduction is that the authors do not state what exactly they are doing in this study to investigate the effects of different muscle parameters on model performance, so it is not clear why they are taking the approach they do in the methods. I recommend that they explain more clearly what they intend to test in the study, and how this study design will do it, in the introduction.

I have several suggestions for figures that I think would make them easier to understand and interpret:

Figure 3 – Because there are so many bars on this graph, it is difficult to see what it happening at individual muscles. Perhaps this figure could be divided into multiple panels for different groups of muscles (like the groups used in Figure 7)?

Figure 5 – Why is the human model in a different color from the australopith models? I think it would be easier to compare between species if they were the same color.

I believe that the raw data are adequately shared and supplied, although I have not tried to apply them in OpenSim so I cannot say for certain.

Experimental design

The authors do not clearly explain why they are using their Au. afarensis model to assess the effects of different muscle parameters on model performance, rather than another previously validated model. The Au. afarensis model is based on a lower limb skeleton that is a composite of elements from multiple individuals, and there is some debate considering the reconstruction of some of these elements (such as the pelvis). Also, the authors must assume that Au. afarensis walked with human-like bipedal kinematics for their simulation, and there is considerable debate about whether this was the case. It is interesting that the australopith model had to be manually manipulated into a different limb posture from that based on the kinematic data in order for the foot to contact the ground. Given the model assumptions, I wonder why the authors did not instead just apply these methods to a human model, such as the Subject03 model used for comparison here. They could compare these different parameters for muscle architecture estimation to the human model and determine how they perform relative to the actual known values. This could provide a robust means of assessing which parameters and models are most accurate.

The authors do not clearly state what criteria they are using to assess the performance of these variants. In results and discussion they seem to suggest that the variants that can withstand higher ground reaction forces perform the best, but the basis for this criterion is never really discussed. Part of the reason that it is unclear as to whether this is a good criterion is because the human model with real muscle data can only withstand 2.2x body weight, and it is not explained why this is not the standard by which performance of the australopith model should be assessed.

I do not see the value of the comparison of the australopith and human model performances. As discussed below, the human model yields results that do not fit with known human muscle activation patterns during walking. Additionally, the assumption of similarity of kinematics between species may not be valid, as evidenced by the manual manipulation of the australopith model that is necessary to run the loading simulation. Additionally, the absence of a medio-lateral component to the ground reaction force vector could have a significant effect on what muscles are actually used during walking.

The authors should provide an explanation as to why the alpha centroid method could only be applied for the isometric approach.

The authors do not clearly explain what the ‘reserve actuators’ are, and why they need to be used for these models.

The authors should explain the use of l to PCSA ratios in the methods. They should also explain what specific cut-off points they use to define a muscle as having a ‘small ratio value’ versus a ‘high ratio value.’

Lines 340-347 – The reason for needing to increase the value of gravitational acceleration should be more clearly explained and justified.

Lines 350-354 – The comparison to previously published models is not clear. What is the difference between the two models? In the results, it seems like the authors only compare the australopith results to those of one human model. The authors describe the body size and demographics of ‘Subject03’ model, but not the ‘FullBodyModel.’

Validity of the findings

Generally, the results for the human model do not seem to match-up well with real human walking data. This model can only support 2.2x body weight, which is well under what humans are known to be able to support on a single limb during dynamic behaviors like running. Additionally, the muscle activation profiles do not match what is typically reported in EMG studies of human walking. For instance, humans activate their lesser gluteals during midstance, but the model showed no lesser gluteal activation. Additionally, the model showed activity in the hip flexors at midstance, but EMG evidence indicates that these muscles are silent in humans at midstance. Similarly, humans do not typically activate their adductor muscles at midstance. Given the mismatch between the human model simulation results and what is known about actual human walking kinetics and muscle activation patterns, it does not seem like this model produces accurate results. This leads me to question the validity of the results from the australopith model variants, at least with regard to muscle activation during actual walking.

Lines 358-366 – This paragraph was confusing to me. Why was the ‘FullBodyModel’ able to withstand so much more force than the ‘Subject03’ model? And why is the ‘Subject03’ model used in subsequent comparisons to the australopith model?

Lines 484-487 – Can the authors explain this statement a little bit more clearly? I think this is critical because this sentence may capture why the human model did not seem to produce biologically realistic results, and it may also speak to the limitations of what we can learn from this static loading approach.

Lines 507-508 – I think the authors should explain this statement more. It is important that they explain what the broader value of their results are to others who are attempting musculoskeletal simulations in different animals, so I think they should give more clear prescriptive advice based on their findings. I also think they should explain the downside of Hill-type approaches. They seem to suggest that the Hill-type variants perform better than the isometric variants, so why not always use Hill-type variants? It would be helpful to spell this out for the readers.

Additional comments

None.

Reviewer 3 ·

Basic reporting

The study is a modeling exercise that compares different methods for estimating muscle architecture in an AL 288-1 model. Human experimental data are used to define model joint and grf positions and then the authors calculate an estimate of the model strength in a human-like mid-stance posture using multiple versions of the model

The manuscript would be improved if the following issues are addressed:

1. The Introduction and Methods text should clearly differentiate between the muscle architecture estimation and the parameterization of the muscle models. As is, the issues are interwoven such that methods involved in each step are difficult to discern, but should be separate.

In particular, the authors should define the ‘centroid’ estimation methods, either mathematically or descriptively. It’s not clear what differences exist among these four approaches for estimating an ‘average’ parameter value. Maybe some of this is in Bishop et al. 2021 to which the authors often refer, but it should be made clearer and elaborated upon to set up the Methods herein. The application in this study is specifically to intraspecific, human data as I understand it.

Along these lines, in addition to Fig 1, the authors might add an associated table with relevant soleus parameter estimates for the fossil model from the different approaches shown side-by-side. This would be helpful in providing readers some sense of how different these approaches are in practice, without having to dig into the supplement.

2. The terms ‘isometric’ and ‘Hill-type’ lack a clear distinction since Hill-type muscles can contract isometrically. The authors might be better served using terms such as ‘static’ and ‘dynamic’, where the contrast is perhaps clearer. But, has anyone used an ‘isometric’ muscle model (as defined by the authors) for static optimization or a dynamic calculation? If not, what is the rational for evaluating it here at all? Some published context for these analyses is needed.

Related to this, more consistency in terminology would really aid the readability of the manuscript. For example, ‘muscle action’, ‘muscle function’, ‘muscle activation’ and ‘muscle recruitment’ are used interchangeably. But, a narrower range of (one or two) well-defined terms is clearer.

Experimental design

3. The Introduction should be revised to set up the Methods and Results more clearly. Data exploration is fine, but the structure of the manuscript comes across as unorganized and meandering in the absence of any clear set of aims. There were instances where methods (ie. ‘model evaluation’) and results (ie. ‘lo/pcsa’ and ‘lo’) appeared unexpectedly, for example. I see no reason that the authors couldn’t define some main questions, aims or even hypotheses in the Introduction to better establish the Methods, Results and Discussion to come later.

Some additional clarifications:

L 59-70: The distinction being drawn between studies that used polygonal modeling and those that do is convoluted. Both approaches are guided by data from comparative living species; both approaches produce a muscle with a line of action with via points denoting the path direction changes. The fact that one provides an estimated muscle volume is great, but immaterial if the goal is studying movement dynamics.

L74-75: Cite an example of a Huxley-based muscle model here. Among other issues, they aren’t used because they are computationally inefficient.

L84-93: In a study of human and fossil hominins models, why are all the previous examples of musculoskeletal modeling from bird and dinosaur models? There are lots of existing human, chimpanzee and macaque musculoskeletal models that seem more directly relevant. Minimally, the authors need to add ‘e.g.’ throughout to note that these citation lists are incomplete.

L94-110: The statement that “… generic, isometric scaling theory does not apply to species which have many geometric differences, as we typically find between fossil specimens (i.e., an australopith) compared to comparative living specimens (i.e. a human)” is incorrect and requires some revision. The author’s preferred citation – Alexander et al. (1981) – does not support this, as it includes mammalian species with marked geometric dissimilarities, but many muscles and muscle groups scaling with isometric expectations. This geometric scaling has also been demonstrated in studies of a range of primates, including apes (see Payne et al; Myatt et al.). This isn’t to say that there are no deviations from isometric expectations of course, but it is untrue that isometric scaling does not apply to geometrically dissimilar samples.

It is also an odd point to argue given the extensive use of geometric scaling to derive the fossil hominin model parameters from normalized human-based parameters (centroid estimates), as outlined in the Methods section.

L 158-160: “… despite some simulated dynamic assessments of movement which do posit this species as a biped”. I’m not sure what the authors are trying to say here since this species is broadly agreed to be a biped, as noted earlier in the sentence.

L 160-162: “These earlier musculoskeletal models, however, were created with simple lines of action and isometric muscle models – limitations which were necessary due to computational constraints at the time”. This is incorrect. Both Nagano and Sellars use Hill-type muscle models and even Wang et al.’s muscle models permit shortening-lengthening in some form, albeit lacking contractile dynamics. All three models include some muscle-tendon paths with via points, similar to the models shown in Fig 2.

L 166-167: “ … this new approach considered the spatial configuration and space-filling for the muscles … thus improving the anatomical fidelity of the muscle paths”. This is a fine approach but improved from what and in what way? What does ‘anatomical fidelity’ mean? In any case, in a fossil taxa where the true parameters are all unknowns how can improvement be assessed?

L 171: Replace “in a quasi-static (inverse) simulation” with what you did here in specific, “static optimization in an approximate human-like midstance walking posture”, or something similar. The current text is unnecessarily opaque.

Section 2.1 Please clarify in the text how many human models were studied and the primary source for that/those models. The authors mention Wiseman, Charles et al and Rajagopal et al as citations for the human model(s), but this should be made clearer. The results in Table 3 include a single subject-specific model, which is the one from Charles et al.?

L 195-198: Please elaborate on the “correction factor” applied to each convex hull approach. How were they derived? How was ‘under-calculation’ of true mass determined for a fossil model?

L 265: The muscle stress citation is incorrect. There are no 300 kN/m^2 measurements of mammalian muscle in the Medler (2002) review. However, direct measurements of muscle stress are available for skeletal muscle fibers from humans, chimpanzees and other mammals that are of direct relevance to this study.

Section 2.3: The authors need to clarify their methods for determining midstance joint posture for their models. First, it is not clear what human model was used (one of the Charles et al. models?) and whether the model was matched to the subject from which the experimental data were collected. Second, what was the ‘cost function’ used in for the OpenSim MOCO inverse kinematics? Third, was this done for the AL 288-1 model, as the first sentence of this section states (“all models … were animated using previously collected motion capture data”)? If so, please elaborate (marker placement, data scaling, etc.).

L 308: The authors note that ‘the applied joint angles were used as a guideline, after which slight manual modifications were made to the hip to position the limb with the foot flat on the ground”. My impression then is that the fossil model was just positioned in similar-ish joint position to the human model. If so, was the inverse kinematics exercise necessary? Please report the joint positions for the human and fossil model.

Validity of the findings

Section 2.4: Not sure what is being evaluated by these analyses. A motivation for this analysis in the Introduction would help. Surprisingly, these are the first results presented. The authors observe that the Rajagopal number is quite a bit larger than that of the Subject03_2019. What does this confirm, exactly?

L 374-375: The authors report that “All variants indicate that the limb was capable of supporting greater vGRF than the humans, ranging between 2.6 … to 3.4 …” These numbers are within the range reported from the two human models in the previous paragraph (2.2 to 3.4), so how does this “indicate” a difference between species?

L 385-387: The authors never define what is a “~small ratio value” and what is a “~high ratio value”. Can this be defined numerically? Otherwise, I’m not sure what the cut-offs are for the two groups. I think it would be more sensible to move this entire explanation and motivation into the Introduction, as mentioned earlier. The Results should just contain the results of the analysis, not the motivation for it and discussion of findings.

L 398-400: The muscle fiber length analysis is also unexpected and not explained. Comes out of left field. Please move the explanation and motivation for this analysis to the Introduction, then just present the results here.

Section 3.3: In looking at the muscle groups, it should be noted somewhere that these are heuristic simplifications and that many muscles will contribute to multiple groups. Even still, it is highly unusual to include the gastrocs in the ‘knee extensors’ and not the ankle plantarflexors, where they have their largest moment-producing capabilities. Note that Figs 5 and 6 sort these muscles differently.

L 494: Define ‘knock-on effects”. I am not sure what the authors mean here.

L 496: The conclusion that “… the exclusion of an elastic tendon and pennation angle produce models that are weaker and require greater muscle recruitment for single limb stance” appears inconsistent with Table 3, where the ‘simple’ 3-D model is weaker than the ‘Hill-type’ 3-D model. Please explain.

---

## Round 0.2 · Major Revisions

Thank you again for submitting your research to PeerJ. Reviewers 2 and 3 from the original submission have evaluated the revised manuscript. Though both appreciate the changes you have made in response to their original comments, they have several additional concerns that should be addressed.

Reviewer 2 articulates some broad concerns with how your study is framed and suggests that you do some more work to articulate the novelty of this study relative to previous work. Along these lines, I wonder if it would be worthwhile, in the Discussion, to note that the different muscle models each carry different assumptions that must be made in the modelling process – particularly when attempting to investigate the morphology and mechanics of extinct taxa. Could it be that “static” models are preferable when otherwise making certain assumptions would be unjustified, or untenable? Just a thought. In addition to these concerns about study novelty, Reviewer 2 details concerns about the criteria for evaluating the different models and the utility of comparing the A. afarensis model to human data.

Review 3’s concerns are more specific, primarily relating to the relevance of certain citations and other claims that are made throughout the manuscript.

Given the number and breadth of the suggestions made by the two reviewers, I am again recommending “Major Revisions”.

In addition to the comments made by the reviewers, I had a few other editorial suggestions/comments:

(Line 51) You don’t define “physiological cross-sectional area” before it’s used here. Maybe just say “cross-sectional area”

(Line 79) “primate forelimb muscle masses” not “primate forelimbs muscle masses”

(Line 82) The citations beginning with Payne[…] should be moved inside the period of the previous sentence.

(Lines 114-115) Define tendon slack length

(Line 130) Extra punctuation at the end of “for instance from velocity-driven (long fibres) to force-driven (short fibres),..”

(Line 330) “Bishop and colleagues (2021)” – is this 2021a or 2021b? Also, conceptually, I have a really difficult time understanding why changing gravity is a valid modelling step. Can some additional explanation be added here?

Reviewer 2 ·

Basic reporting

See additional comments.

Experimental design

No comment.

Validity of the findings

See additional comments.

Additional comments

I appreciate the changes that the authors have made to improve this manuscript. Adding in medio-lateral forces to the model and simplifying the analysis involving the human model are both big improvements. I also appreciate the attempt to clarify the purpose of the study. However, I think there are several major lingering issues: the novel advances of the study still need to be more clearly stated, the validity of the model performance benchmark is questionable, as is the human versus australopith comparison, and the authors should explain the modeling jargon in the methods more clearly. These comments are described in more detail below.

I think the authors still need to do a better job of explaining the novelty of the study, as it took me some time to figure this out, even after the revisions. From what I can ascertain, previous studies like the one by Demuth et al have looked at different methods to estimate muscle architecture in extinct taxa and observed the effects on model output. What appears to set the current study apart from these previous studies is that in addition to this approach, it also compares outputs from models with ‘static’ vs. ‘dynamic’ (i.e. ‘no tendon’ vs. ‘tendon’) muscle models. However, what isn’t clear to me is what the advance of this study is over studies which have looked at the effects of muscle architecture parameters on human models (e.g. Kramer et al 2022 and Charles et al 2022). Can the authors please clarify what we learn from this study that we have not already learned from studies of human models? I would also appreciate if the authors would explain more clearly why this should be done in a fossil hominin model that can’t be validated, rather than just using human models that can be better validated.

One lingering issue is that the authors still do not lay out good criteria for evaluating the quality of the different models. They suggest that australopith models should be able to withstand as much or more force (relative to body weight) than the human model. I see a few related problems with this. First, why is this a good criterion for model performance? Why should better models be able to withstand ever increasing amounts of force? I would guess that there’s an optimum that is not just related to withstanding higher amounts of force, and it’s possible that it’s a lower amount of force. Second, I don’t agree with the logic that australopiths should be able to withstand higher forces than humans. The cited Biewener 1989 paper states that smaller animals need to exert higher muscle forces because of their more flexed limb postures, but the human and australopith models used in this study had nearly identical limb postures (see comment below). Third, during walking gaits the body typically only needs to sustain well below 1.5 body weights, so why is inability to sustain 2.2 body weights relevant in walking? Fourth, the authors acknowledge that there are many assumptions about the australopith joint postures used in their model simulation, and that australopiths may have used somewhat different midstance postures, so the comparison to humans already rests on some large assumptions that may not be true. Given all these things, it’s hard for me to agree that some models clearly ‘underperform’ relative to others, since the performance benchmark seems a bit arbitrary to me. At the very least, I think the authors should discuss more of the caveats to using 2.2 body weights as a performance benchmark, and be open to the possibility that lower values do not necessarily mean underperforming models.

Another lingering issue related to those described above is that the authors still do not provide any clear reasons as to why one would want to use a static model instead of a dynamic one. There are no examples in the discussion of where static models would be preferred, so why not conclude that one should always use dynamic models? One of the other reviewers raised this question, and I think it’s an important one for justifying the purpose of this study. The authors cite several previous studies that used static models, so perhaps those studies provide examples of cases where static models are preferable.

One more important issue I see with this manuscript is that the authors still make comparisons between humans and australopiths (i.e. study aim 5), despite numerous disclaimers that such comparisons are not a goal of the study (e.g. lines 155-158, 178-180, 590-597, 601-605). They have added in much language to acknowledge all the assumptions about similarity in joint postures between humans and australopiths that may not be true, as well as static optimization rules that obviate the output of realistic gait variables. However, they still include comparisons between the two species and make broad conclusions about australopith locomotion (lines 474-507, 597-600). I don’t think they can devote so much text to stating that the study objective is not to compare species, and giving all the reasons why such a comparison might not be reliable, and then still retain a comparison between species. It seems to contradict much that is said elsewhere in the paper and makes it sound like they don’t appreciate all the limits of the modeling approach described in this manuscript in terms of attempting to reconstruct australopith gait. Personally, given all their caveats, I don’t see the value of the between species comparison.


Other issues:

Throughout the paper (e.g. line 161) the authors use the descriptor ‘upright’ to refer to walking with relatively extended lower limb joints. I think this term is far too vague, and encourage the authors to use more specific language, such as ‘relatively extended lower limb joint postures’ or ‘human-like lower limb joint postures.’ I think the term ‘upright’ is meaningless in and of itself.

Line 200 – There are several AL-333 specimens, so I think the authors should describe specifically which ones they used.

Lines 323-329 – Generally, I would like the authors to explain more of the modeling jargon. I think they should explain more clearly what dynamic congruity and residual actuators are.

Lines 330-33 – As with the comment above, it is not clear to me what ‘dynamic consistency’ or ‘force equilibrium’ are. Can this be explained in layman’s terms?

Lines 357-359 – I don’t think this is quite what the Biewener study says. That article does not state that smaller animals can exert relatively larger forces than larger animals. Rather, it states that smaller animals tend to need to exert larger muscle forces during gait because of their use of more flexed limb postures than larger animals. However, why would this be relevant in the present study, since one of the assumptions of these models is that humans and australopiths used similar lower limb postures during gait?

Reviewer 3 ·

Basic reporting

Thanks to the authors for their attention to my earlier comments. The manuscript is improved from the initial submission. I suggest the following additional points for consideration:

“Australopith” is a shorthand for a group that includes multiple species of Australopithecus, but only afarensis is modeled here so the authors should just refer to the species or model in specific.

The authors need to be more conscientious in their citations, ensuring that the papers they are citing do or say the things that they are citing them for. A few illustrative examples of problematic citations, …

a. Payne et al. (2006: 271) report that, “Proximal limb mean fascicle length scaled to (body mass)^0.30 and distal limb to (body mass)^0.34. Thus, geometric scaling of the data is supported.” They do not present scaling exponents for PCSA. Yet, the authors use this paper to say, “In primates, … hind limb fiber lengths and physiological cross-sectional areas often scale with positive allometry (Payne et al. 2006)”. This is incorrect and should be revised.

b. DeSilva et al. (2018) refers to a juvenile specimen that is from a different locality than AL 333. So, the citation appears incorrect, and it is unclear what specimen(s) were used for the AL 288-1 foot or how they were scaled to fit. AL 333 represents multiple individuals. There are foot elements from this site, but not all AL 333 material is associated with the foot and not all foot bones are associated with the same individual.

b. Zhang and DeSilva (2018) is not a musculoskeletal model. It is a skeletal model where synthetic data were used to create a walking animation. It is not a forward dynamic simulation or mechanics-based study.

L67 There is a new afarensis model in AJBA by O'Neill et al. that the authors can add to this list.

L68 “Architectural estimates are typically scaled from living species with analogous muscle morphology (Nagano et al., 2005) parsimoniously conserved or ‘phylogenetically bracketed’ … ” I think these are all the same approach, but it sounds like the authors are drawing a distinction between them, but I don’t follow what the distinction is.

The authors should also note that this approach would suggest the consideration of non-human-based muscle model estimates, since this is not a model of a human species.

L91 “Bishop and colleagues (2021a) developed a scaling method …” Given the proceeding paragraph, it is notable that the ‘normalization’ of fiber length and muscle mass is based on isometric scaling. This seems perfectly fine to me, but this should be made clear in the text. Something like, “… a method based on isometric scaling of muscle architecture ...”.

L112 Please add citations to studies of extinct species that "ignored the role of the tendon", to reinforce this point.

L136. This is a rather limited presentation of the large number of studies that have investigated effects of architectural properties on modeling and simulation results, especially in human models. The authors might make clear that the effects of muscle architectural properties on Hill-type muscle models outputs have been probed extensively.

In contrast, Kramer et al. (2022) is unusual in that they don’t use a Hill-type muscle model, so the results/implications are likely much more circumscribed.

L193 What are the comparative dissection data?

L261 The authors describe their work as producing “seven unique … models” and “seven different estimates of architecture per muscle” of a single model. Since the skeleton and muscle-tendon paths are consistent across conditions, this latter description is clearer for the reader and I suggest the authors use it throughout.

L291 Please articulate why this is described as a “quasi-inverse” and not “inverse” simulation.

L308 The statement "these produced a set of joint angles across an entire stride (Table 2)" is off a bit, since Table 2 is for a single time point in a stride, not an entire stride. I'd drop the reference or change the text.

L313 Add a reference to Table 2 here.

L353 Is there any quantitative assessment with the published EMG data? If not, state how are they compared. Also, it is useful to point out that any such comparisons would be at a single midstance timepoint in a stride, not over the entire curve.

L479 The “foot plantarflexors” should be clarified. I think the authors are referring to what are called the deep plantarflexors (EDL, EHL, TP) in humans. This is differentiated from the superficial plantarflexors (LG, MG, Sol) which also plantarflex the foot/ankle, but do not travel along the plantar surface of the foot. Foot plantarflexors is too vauge.

L500 The authors need to be clear that these are estimates of muscle force-length positions at a single time point and not over a walking stride. Also, they are based on artificially prescribed joint positions. These results are not visualized (which is fine), and I am concerned that this may be a point of confusion for some readers as they go through the manuscript ... even if the authors are clear on this point.

L604 Only Falisse et al. 2019 and Nagano et al. 2005 are predictive simulations. Either the text or the citations need to be amended here.

Experimental design

1. The discussion of ‘static muscles’ could use some additional clarifications. The text states that static muscles assume rigid tendons and exclude force-length-velocity properties in all cases, whereas the dynamic muscle assumes compliant tendons. Do the dynamic muscles also include force-length-velocity properties?

In general, I think a bit more detail on how the Millard muscle models were modified between the static and dynamic conditions would be helpful. How was the tendon made rigid and FLV properties removed? Also, for the human model, is it stated somewhere whether the tendon compliance and FLV properties were retained?

I also think a bit more published context of when a ‘static’ and ‘dynamic’ muscles would be used is needed. The authors provide citations to studies that use rigid tendons but did these studies also exclude FLV properties? If not, then make clear that the ‘static’ condition is simply a theoretical exercise without an applied context in the literature.

2. Related to this, in looking at Fig 1, there is a substantial difference between the estimated parameters for the soleus between the ‘static-muscle’ model and the ‘dynamic-muscle’ model, but I do not understand why this would be the case. The authors state that these two conditions differ in the inclusion of tendon compliance and FLV properties.

However, Fmax, for example, is only dependent on the architectural properties of the muscle (mass, fiber length, pennation angle), so why would inclusion of tendon compliance and FLV properties in the ‘dynamic muscle model’ double or halve the parameter estimates from the ‘static muscle’ model? I think I’m missing something here in the description or some clarification in the text is required.

This might be a more general issue if this is a consistent trend between the ‘static’ and ‘dynamic’ muscle model datasets. I’ll also add that, based on this figure, the ‘static muscle’ model also excludes pennation angle, which isn’t mention in the first paragraph of section 2.2.

3. The additional citations are helpful for muscle stress -- although the authors have still missed some muscle contractile data for nonhuman primates (e.g. chimpanzees, macaques), which would seem more relevant here.

I understand the authors point about muscle stress, but it is worth mentioning that the specific stress values do very much matter for the absolute magnitude of the simulation result. If you double this value, as in some human models, the SO with “fail” at much higher number, even if the relative performance among the architecture estimation conditions is the same.

So, the fact that some conditions cannot support >1.8 times body weights is dependent on the 30 N/cm^2 number. My concern is that some readers will fixate on the absolute values, even if only the comparisons matter.

Validity of the findings

L598 The authors state that “… the inference that this individual could stand and locomote on an upright limb ...”, but these results are not predictive of standing or locomotion. I think the forces and moment required in a static simulation of midstance are so low that I’d be surprised that any primate on two limbs couldn’t do it, even if they don’t all habitually walk or stand upright.

As Monty Python has demonstrated, humans can stand and walk in many ways, but we prefer to use just a few of these possibilities. And why we prefer those few is a very different matter than what is addressed here. So, I would argue that these results are entirely agnostic on the kinematics and forces in australopiths (i.e. standing or locomoting on an upright limb), not just on the frequency and efficiencies of their locomotion.

---

## Round 0.3 · Minor Revisions

Thank you for the thorough job you did in responding to my comments and the remaining concerns from the two external reviewers. In reviewing the latest revision, I found only a few minor areas that require typographical correction or additional clarification. Once these comments are addressed, I would be happy to recommend Acceptance. All of the line numbers below refer to the submitted PDF, not the Tracked Changes document.

(Lines 69-70) Please cite the O’Neill et al, 2023 and Wiseman, 2023 studies at the end of this first sentence in the paragraph.

(Line 340) Thank you for the thorough explanation on why it is necessary to heuristically increase gravity in these static simulations. To further clarify this necessity to the reader, I suggest adding “static” in front of “simulations” in Line 340, so the edited sentence would read, “Following Bishop and colleagues (2021a), gravity was increased accordingly relative to m_body during the static simulations to produce net force balance and to help achieve dynamic consistency.”.

(Line 448) There are two question marks after “levels)”.

(Line 530) I think you should specify that the human EMG data referred to here were sampled during walking/running (rather than the static situation modelled here); something like “Nevertheless, the muscles in the human model overall matched well with published EMG data sampled during human locomotion (i.e., Perry and Burnfield, 2010, Van Criekinge et al., 2018, Cappellini et al., 2006, Wall-Scheffler et al., 2010).”.

---

## Round 0.4 · accepted · Accept

Thank you for your thorough and thoughtful responses during the review process. I have reviewed your edits in response to the final sets of recommendations, and I'm happy to now recommend acceptance of this manuscript.